# UNSUPERVISED LEARNING OF FACIAL ATTRIBUTE REPRESENTATIONS USING STYLEGAN

## ABSTRACT

Facial attributes (e.g., gender, age) encompass important social cues and play a pivotal role in computer vision. While supervised methods have dominated facial attribute analysis, they often require large annotated datasets, which are costly and time-consuming to create. In this work, we circumvent this limitation by proposing a novel unsupervised learning framework that leverages StyleGAN to learn rich and disentangled facial attribute representations. Specifically, unlike prior methods that rely on labeled datasets or supervised techniques, our approach exploits the unique inductive bias of StyleGAN, namely Hierarchical Feature Modulation, to automatically discover semantically meaningful representations of facial attributes. This inductive bias enables StyleGAN to generate disentangled and interpretable facial attribute features at different layers, benefiting a variety of downstream tasks. To leverage StyleGAN representations, we employ GAN inversion methods to represent input images as StyleGAN features and propose a simple yet effective feature reduction method based on mutual information to improve the effectiveness and efficiency of the learned representations. Extensive experiments in few-shot facial attribute analysis tasks, including clustering, classification, and facial attribute annotation demonstrate the effectiveness of our approach.

## 1 INTRODUCTION

Facial attributes, such as gender, age, and the presence of accessories like glasses, are crucial social cues that play a pivotal role in human perception and interaction. In computer vision, the accurate analysis and representation of these attributes have far-reaching implications for various applications (Kortli et al., 2020; Zheng et al., 2022; Narayan et al., 2024).

Traditionally, supervised learning methods have dominated the landscape of facial attribute analysis, achieving remarkable performance in tasks such as attribute classification and detection (Li et al., 2022; Qin et al., 2023; Kuprashevich & Tolstykh, 2023). Recently, various methods based on Masked Autoencoders (MAEs) (He et al., 2022) have significantly advanced the field by introducing (unsupervised) representation learning to facial attribute analysis to improve classification accuracy, facilitate multi-modal learning, reduce computational costs, etc. Notable examples include ABAW5 (Zhang et al., 2023), MCM (Zhang et al., 2024), MAE-DFER (Sun et al., 2023), and MARLIN (Cai et al., 2023). However, all these approaches rely on large, meticulously annotated datasets, which are both costly and time-consuming to create. This dependence on labeled data presents a significant bottleneck in advancing the field and limits the scalability of facial attribute analysis to new domains or attributes.

In this paper, we address this challenge by proposing a novel unsupervised learning framework that leverages the generative power and unique architecture of StyleGAN (Karras et al., 2020b; 2021) to learn rich and disentangled facial attribute representations. Our approach represents a paradigm shift in facial attribute analysis, moving away from the reliance on labeled datasets and supervised techniques towards a more flexible and scalable few-shot methodology. At the core of our framework is the exploitation of StyleGAN's distinctive inductive bias, specifically its Hierarchical Feature Modulation technique. This architectural feature enables StyleGAN to generate highly realistic facial images while maintaining fine-grained control over various attributes (Viazovetskyi et al., 2020; Wu et al., 2021). As shown in Fig. 1, our key insight is that this inductive bias also allows StyleGAN to automatically discover and represent semantically meaningful facial attributes in a

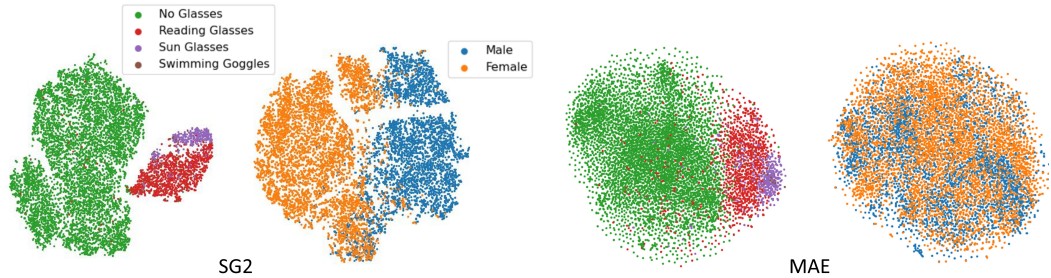

Figure 1: t-SNE visualization of the representations learned by StyleGAN2 (SG2) (Karras et al., 2020b) and Masked Autoencoder (MAE) (He et al., 2022) of 10,000 images from the FFHQ dataset (Karras et al., 2019) for glasses and gender attributes.

disentangled manner across different layers of its generator network. To leverage the representations learned by StyleGAN, we employ GAN inversion methods (Tov et al., 2021) to map input images to StyleGAN features. We then propose a simple yet effective feature reduction method based on mutual information to enhance both the effectiveness and efficiency of the learned representations, facilitating downstream tasks. Extensive experiments in few-shot downstream tasks, including facial attributes clustering, classification, and annotation, demonstrate the effectiveness of our approach. Our main contributions include:

- We propose a novel StyleGAN-based facial attribute representation learning framework, which learns highly disentangled and distinguishable features in an unsupervised manner.

- To the best of our knowledge, ours is the first approach to enable *few-shot* downstream tasks for facial attribute analysis, including clustering, classification, and annotation, achieving significant improvements over state-of-the-art methods.

- As a valuable by-product, we manually annotated the AFHQ-Wild dataset (Karras et al., 2020a) and will release the labeled dataset upon acceptance.

## 2 RELATED WORK

**Supervised Facial Attribute Classification.** Supervised facial attribute classification has been widely studied, with deep learning models like CNNs (Kalayeh et al., 2017) traditionally used to identify attributes such as age, gender, and expressions. Recent advancements, particularly transformer-based architectures, have further improved performance across multiple facial attribute analysis tasks. For instance, the Label2Label (Li et al., 2022) framework addresses multi-attribute learning as a sequence generation task, using a language modeling approach to better capture relationships between attributes, enhancing classification accuracy. Similarly, SwinFace (Qin et al., 2023), a transformer-based architecture, adopts multi-task learning to handle face recognition, expression, age, and attribute estimation in one framework, leveraging transformers' hierarchical structure to improve overall performance. Meanwhile, Mivolo (Kuprashevich & Tolstykh, 2023) uses a multi-input transformer to focus on age and gender classification, highlighting the value of integrating multiple facial cues for more accurate predictions. Nevertheless, these methods have encountered a bottleneck of facial image representations, prompting recent approaches to shift towards representation learning, particularly those based on Masked Autoencoders (MAEs) (He et al., 2022).

**MAEs-based Facial Representation Learning.** Masked Autoencoders (MAEs) (He et al., 2022) have introduced promising methods for facial attribute image representation. These techniques leverage the power of self-supervised learning to extract features from facial images without requiring labeled data. These self-supervised architecture enables a variety of facial analysis tasks without requiring labelled data. For instance, ABAW5 (Zhang et al., 2023) explores the use of MAEs in affective analysis, showing that MAEs can efficiently learn representations from masked portions of facial images. Similarly, MCM (Zhang et al., 2024) proposes a method that combines channel and spatial masking to enhance facial action unit detection, enabling the network to focus on subtle facial movements, but often requiring more complex architectures for deeper facial understanding. For

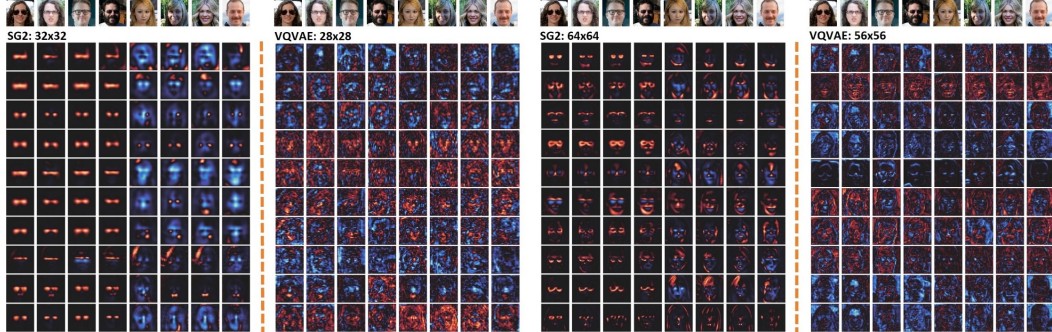

Figure 2: Feature maps of the StyleGAN2 (SG2) (Karras et al., 2020b) and VQVAE (Rombach et al., 2021) models, both of them are pretrained on FFHQ. Top row: input images from the FFHQ dataset (Karras et al., 2019), 4 with glasses and 4 without.

dynamic facial expression recognition, MAE-DFER (Sun et al., 2023) applies MAEs to time-variant facial data. It demonstrates the ability to learn temporal features for dynamic expressions, yet the approach struggles with disentangling individual facial attributes effectively. Similarly, MARLIN (Cai et al., 2023) extends this to video sequences, highlighting MAEs' capability in temporal learning but also facing challenges in isolating distinct facial attributes across frames, limiting its effectiveness in facial image representation. Despite their success, MAE-based methods lack inductive biases tailored to facial attribute analysis, leading to less disentangled and distinguishable representations. Thus, they continue to rely on fully-labeled datasets, which are costly and time-consuming to create.

In this work, we address this challenge by introducing a novel unsupervised facial attribute representation learning method based on StyleGAN (Karras et al., 2019; 2020b), leveraging its unique inductive bias (i.e., Hierarchical Feature Modulation). This allows StyleGAN to automatically learn disentangled and distinguishable representations, facilitating few-shot downstream tasks. To the best of our knowledge, this is the first method to successfully achieve such results.

## 3 FACIAL ATTRIBUTE REPRESENTATIONS LEARNED BY STYLEGAN

Our key insight is that the unique inductive bias of StyleGAN (*i.e.*, Hierarchical Feature Modulation) enables the automatic discovery of semantically meaningful representations of facial attributes.

**Facial Features Learned via Hierarchical Feature Modulation.** StyleGAN introduced a groundbreaking approach to image generation through its Hierarchical Feature Modulation. Similar to previous works, this method employs a pyramid of convolutional blocks that synthesize images in a coarse-to-fine manner. However, at each layer, the feature maps are modulated via adaptive instance normalization (AdaIN), which scales and shifts them based on style information derived from a latent vector. The distinct advantage of this approach is that it *directly* (i.e., no need to propagate the style vector through layers) modulates each feature map *independently* (i.e., with unique modulation parameters) based on the input style, allowing for more disentangled and fine-grained control. In this work, we observed that this strategy not only improves image generation and editing (Abdal et al., 2019; 2020; Alaluf et al., 2022), but also automatically learns a semantically meaningful feature representation of facial attributes.

**Qualitative Illustration.** As an intuitive illustration, we visualize and compare selected feature maps from StyleGAN2 (SG2) (Karras et al., 2020b) and VQVAE (Rombach et al., 2021) pretrained on the FFHQ dataset using 8 input facial images obtained from the FFHQ dataset: 4 with *glasses* and 4 without (Fig. 2). These images are fed into SG2 to obtain their representations by GAN inversion (Tov et al., 2021). It can be observed that SG2 successfully capture the *glasses* attribute in each of their extracted feature maps, whereas most feature maps in VQVAE fail to clearly exhibit the texture of *glasses*. In addition, there are notable differences in their representation quality. StyleGAN demonstrates superior disentanglement of the *glasses* feature, isolating it more effectively from other facial attributes. In contrast, VQVAE's feature maps exhibit lower disentanglement, mixing substantial irrelevant features such as hair, mouth, and cheeks. This observation demonstrates

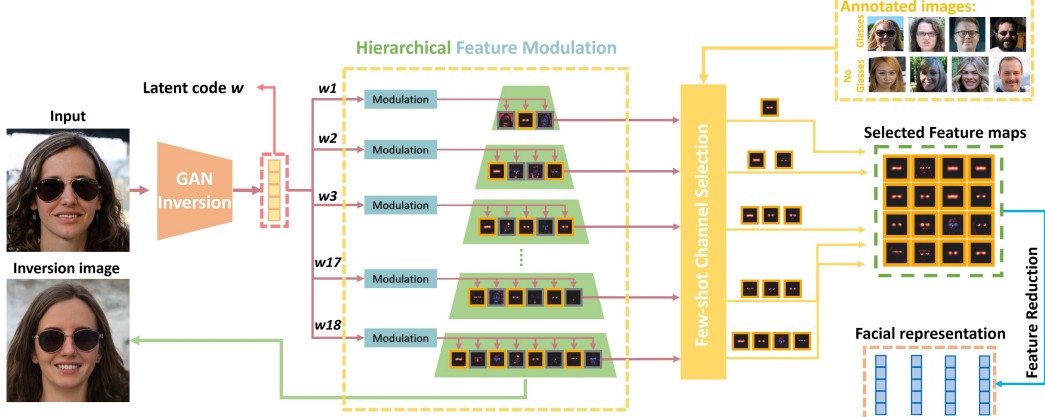

Figure 3: Illustration of the facial attribute representations learned by our method. To obtain these representations, we input an image into the StyleGAN generator using GAN inversion techniques , producing a latent code $w$ that reconstructs the input image. Thanks to StyleGAN's unique Hierarchical Feature Modulation strategy, the learned features are disentangled and distinguishable. While these representations are already valuable, we further optimize them for specific facial attributes through our few-shot channel selection and feature reduction method, enhancing both their effectiveness and efficiency.

that StyleGAN's architecture leads to a more focused and representative encoding of the *glasses* attribute.

**Quantitative Justification.** To quantitatively justify our observation and demonstrate its generality across facial attributes, models, and layers, we propose to use the *difference* between i) intra-class mutual information (IntraMI) and ii) inter-class mutual information (InterMI) as a metric to measure the disentanglement and distinguishability of the representations learned. Specifically, given two feature maps $X$ and $Y$, mutual information(MI) measures how related one feature map is to the other, indicating their similarity. Its formal definition is as:

$$MI(X;Y) = \sum_{x \in X} \sum_{y \in Y} p(x,y) \log \frac{p(x,y)}{p(x)p(y)} \tag{1}$$

where $p(x,y)$ is the joint probability distribution of $X$ and $Y$, and $p(x)$ and $p(y)$ are the marginal probability distributions of $X$ and $Y$, respectively. IntraMI measures mutual information within the same class, while InterMI measures it across different classes. In short, high (IntraMI - InterMI) is desirable, as it indicates high similarity within a facial attribute class and low similarity between different facial attribute classes (e.g., glasses vs. non-glasses), respectively. Specifically, we calculate the average IntraMI and InterMI for each layer of 4 models (SG2, VGG16, VQVAE (van den Oord et al., 2017) and Inceptionv3 (Szegedy et al., 2016)) for each of the 4 facial attributes (glasses, gender, age (man) and age (woman)), respectively. For the 4 facial attributes, we use the labels from the FFHQ-Features-Dataset (Karras et al., 2019) and specify i) two classes for glasses (*glasses* vs. *non-glasses*); ii) two classes for gender (*male* vs. *female*); iii) three classes for age (man) and age (woman), i.e., child (age<10), adult (10≤age<60), and senior (60≤age), respectively. For each class, we select 4 corresponding images from the FFHQ dataset. Similarly, the input images are fed into SG2 by GAN inversion (Tov et al., 2021). Then, for IntraMI, we compute its mean by first averaging the IntraMI between feature maps of the same channel among all possible pairs of the 4 images in a class, and then averaging across all feature maps in a layer; for InterMI, we compute its mean by first averaging the InterMI between feature maps of the same channel among all possible image pairs of different classes for an attribute, and then averaging across all feature maps in a layer. As Fig. 4 shows, SG2 produces much higher (IntraMI - InterMI) than other models, especially in low-resolution layers. This supports our observation that SG2's Hierarchical Feature Modulation technique effectively promotes the learning of disentangled and distinguishable facial attribute features, particularly in the low-resolution layers.

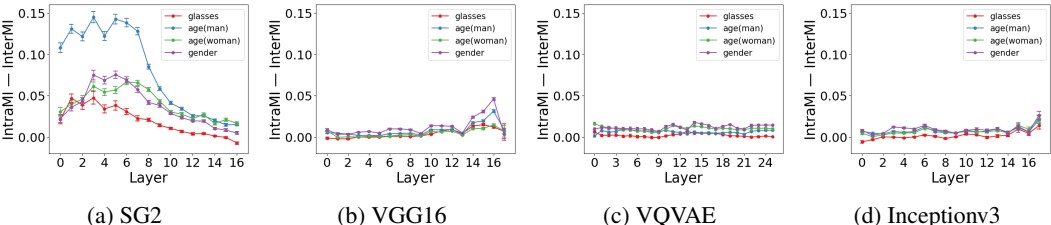

Figure 4: The *difference* between intra-class mutual information (IntraMI) and inter-class mutual information (InterMI) for each layer of the SG2, VGG16, VQVAE and Inceptionv3 models for 4 facial attributes (glasses, gender, age (man), age (woman)). The error bars are generated from repeated experiments using different selections of input images.

## 4  EFFECTIVE AND EFFICIENT REPRESENTATIONS VIA FEATURE REDUCTION

As shown in Fig. 4, while StyleGAN's learned representations are useful, their effectiveness varies significantly across layers and channels, leaving room for improvement. To this end, we propose a novel two-step feature reduction strategy to improve the effectiveness and efficiency of StyleGAN-learned representations as follows.

**Step 1. Few-shot Channel Selection.** Given that the most effective feature channels vary across different facial attributes, we use a small set of labeled input images to identify the most relevant channels. Specifically, as mentioned above, given a facial attribute (e.g., glasses), we select 8 images from the FFHQ dataset: 4 with glasses and 4 without, and obtain their StyleGAN feature maps via GAN inversion. Leveraging the effectiveness of mutual information in assessing disentanglement and distinguishability (Sec. 3), let $a$ be the intra-class mutual information (IntraMI) and $r$ be the inter-class mutual information (InterMI) of a feature map, respectively, we define the distinguishability of the corresponding channel $c$ as:

$$d(c) = \frac{a(c)}{r(c)} \qquad (2)$$

Then, we select the most effective channels as the set $C$:

$$C = \{c_1, c_2, ..., c_n\} \qquad (3)$$

where $d(c_i)$ $(1 < i < n)$ is among the top 10 of its corresponding layer.

**Step 2. Feature Reduction via Max Pooling.** To further reduce the dimensionality of the selected channel features $f(c)$, we apply max pooling and obtain the final representation of an input image $I$ by concatenating the pooled features $\hat{f}$:

$$R(I) = [\hat{f}_1, \hat{f}_2, \ldots, \hat{f}_n], \hat{f}_i = \text{MaxPool}(f(c_i)) \qquad (4)$$

## 5  EXPERIMENTS

### 5.1  EXPERIMENTAL SETUP

**Datasets.** We used the FFHQ-Features (FFHQ-features, 2020) and CelebA (Liu et al., 2015) datasets in our main experiments. To demonstrate the generalizability of the proposed method, we further tested it on the AFHQ-Wild dataset (Karras et al., 2020a). Notably, we manually annotated AFHQ-Wild as it lacks labels.

**Pre-trained Models.** We use StyleGAN2 pretrained on the FFHQ and AFHQ-Wild datasets; MAE pretrained on the FFHQ dataset; VGG16 pretrained on the ImageNet dataset Deng et al. (2009); VQVAE pretrained on the FFHQ and MAE dataset; Inceptionv3 pretrained on the ImageNet dataset; in our experiments.

**Downstream Tasks.** To demonstrate the effectiveness of the representations learned by our method, we conduct experiments on three novel few-shot downstream tasks for facial attribute analysis: clustering, annotation, and classification. For all downstream tasks, we use the representations improved by the method described in Sec. 4.

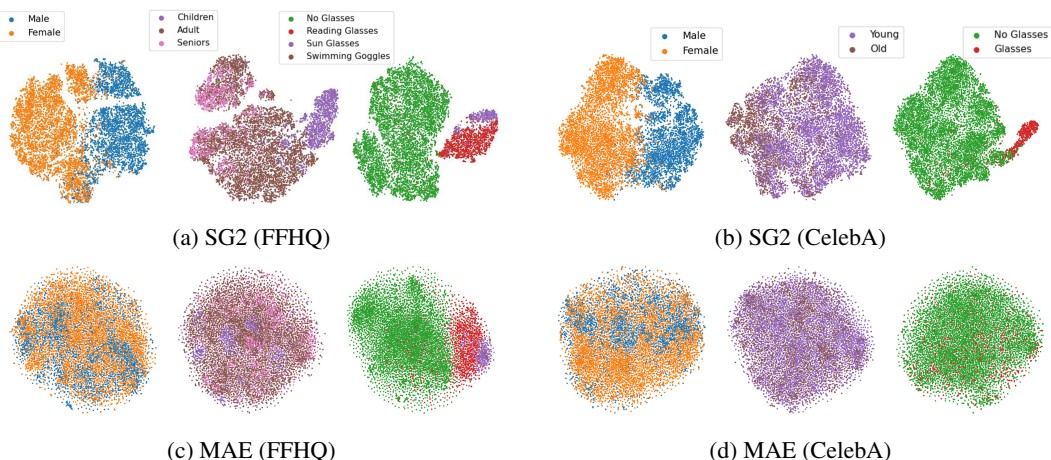

Figure 5: Visualization of representations learned by StyleGAN2 (SG2), VGG16, and Masked Autoencoder (MAE) for three facial attributes (gender, age, glasses) using t-SNE. (a), (c), and (e) show results for SG2 and MAE based on 10,000 images from the FFHQ dataset, while (b), (d), and (f) show results from the CelebA dataset, also consisting of 10,000 images.

Our experiments were conducted on an NVIDIA RTX 3080 GPU.

## 5.2 FEW-SHOT CLUSTERING

Leveraging the disentanglement and distinguishability of our StyleGAN-based facial attribute representations, we achieve effective clustering by simply applying $k$-means, where the number of clusters $k$ is determined based on the few-shot input data. As Table 1 shows, our method significantly outperforms competing methods in few-shot clustering.

| Representation | Glasses | | | Gender | | | Age | | |
|---|---|---|---|---|---|---|---|---|---|
| | ACC | NMI | ARI | ACC | NMI | ARI | ACC | NMI | ARI |
| SG2 (FFHQ) | **97.66** | **80.06** | **90.94** | **87.67** | **52.51** | **57.75** | **79.06** | **35.48** | **35.62** |
| MAE (FFHQ) | 93.67 | 57.07 | 73.87 | 53.98 | 0 | 0 | 66.92 | 0 | 0 |
| SG2 (CelebA) | **98.07** | **65.14** | **80.5**8 | **94.33** | **68.87** | **78.57** | 77.47 | 0 | 0 |
| MAE (CelebA) | 93.67 | 0 | 0 | 58.42 | 0 | 0 | 77.47 | 0 | 0 |

Table 1: Facial attribute clustering results using representations learned by SG2 and MAE across different attributes (glasses, gender, age) on the FFHQ and CelebA datasets. Metrics include Accuracy (ACC), Normalized Mutual Information (NMI), and Adjusted Rand Index (ARI). **Note**: the *zero* and *repeated* values indicate a failure mode where all samples are assigned to a single cluster with the same label.

**T-SNE Visualization.** To more intuitively understand the superiority of our representations, we visualize them alongside other competing methods using t-SNE (Van der Maaten & Hinton, 2008). As shown in Fig. 5, our SG2-based representation exhibits clear class boundaries across all three attributes (gender, age, glasses), in contrast to VGG16 and MAE. This provides intuitive insights into how our representations enhance few-shot clustering performance.

## 5.3 FEW-SHOT ANNOTATION

Our few-shot annotation process begins by converting input images into their StyleGAN representations using GAN inversion and our method. These representations are then clustered into $k$ clusters using $k$-means (Sec. 5.2), where $k$ is a *user-specified parameter*. For each cluster, a small number of $n$ images are randomly selected for manual annotation. The most frequently occurring class among the annotations is assigned as the final label for the entire cluster. Fig. 6 shows how annotation

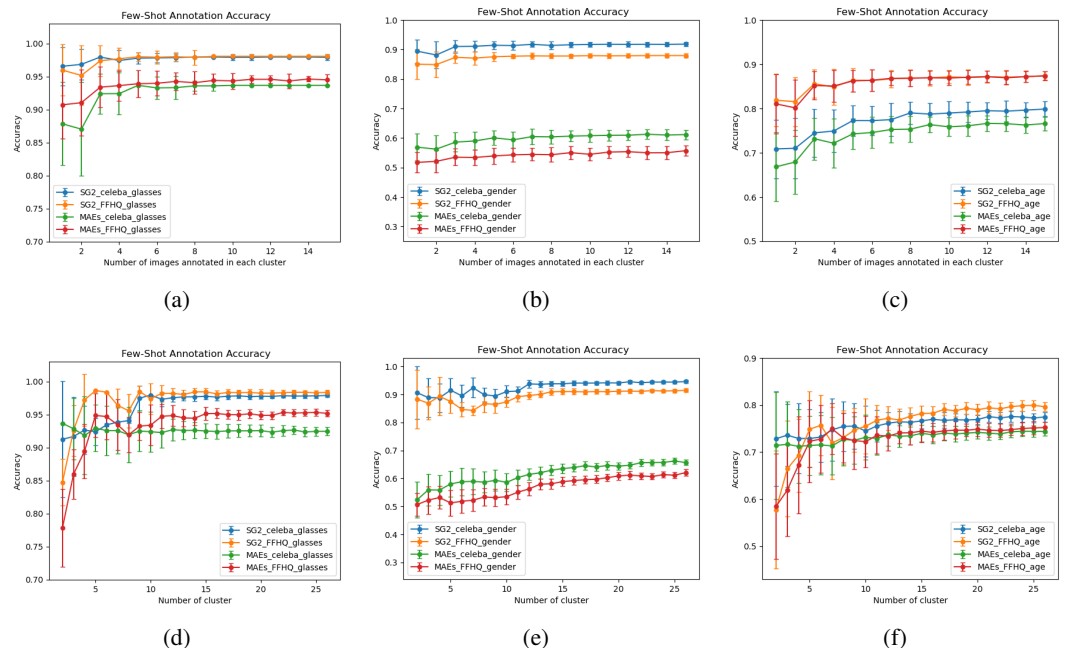

Figure 6: Few-shot annotation accuracy comparison using StyleGAN2 (SG2) and Masked Autoencoder (MAE) representations for three facial attributes: glasses (a)(d), gender (b)(e), and age (c)(f). Among them, (a)(b)(c) illustrate how accuracy changes with the number of annotated images per cluster $n$, while (d)(e)(f) show accuracy variation with the number of clusters $k$. Accuracy is computed by comparing annotated labels with ground truth on 10,000 images from the FFHQ and CelebA datasets, respectively. It can be observed that SG2-based representations consistently outperform MAE-based ones across all attributes and datasets, with the most significant improvement observed in the *gender* attribute, where SG2 achieves nearly a 40% boost in accuracy. Notably, for the *glasses* and *gender* attributes, annotating just 5 images per cluster results in over 90% accuracy.

accuracy changes against the choices of $k$ and $n$. It can be observed that the proposed method outperforms MAE in almost all cases, demonstrating its effectiveness. In addition, in Table 2, we show how annotation accuracy changes against the total number of annotated images. Similarly, it can be observed that our SG2-based representations outperform MAE-based ones by a large margin.

## 5.4 FEW-SHOT CLASSIFICATION

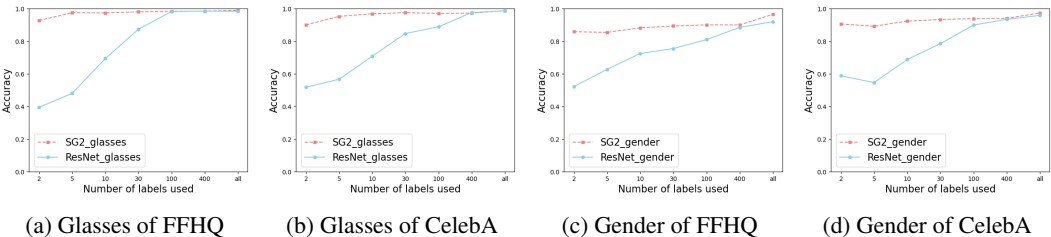

(a) Glasses of FFHQ     (b) Glasses of CelebA     (c) Gender of FFHQ     (d) Gender of CelebA

Figure 7: Few-shot classification comparison of ResNet-18 against our StyleGAN2(SG2) based few-shot annotation across two datasets FFHQ and CelebA for two attributes: glasses (a)(b) and gender(c)(d).

Our few-shot classification is implemented by training a simple 3-layer multi-layer perceptron using the images and labels obtained from our few-shot annotation method (Sec. 5.3). As shown in Table 2, our SG2-based representations outperform MAE-based ones by a large margin and achieve satisfactory results with only 5 labeled images.

**Data Requirement for Supervised Facial Attribute Classification.** To establish the necessity of unsupervised methods for few-shot classification, we trained a ResNet-18 model with varying number of labels to evaluate the amount of data required for an effective facial attribute classifier. Fig. 7 shows that for attribute 'Glasses' (a)(b), while using large number of labels($> 100$) allow both of ResNet and SG2 to achieve accuracy comparable to using full set labels, ResNet presents significantly lower accuracy than SG2 in few-shot setting(fewer than 30 labels). Attribute 'Gender' results (c)(d) similarly demonstrate that a small number of labels is insufficient to achieve good performance in facial attribute classification.

Table 2: Few-shot annotation accuracy against the total number of annotated images. For most cases, 3 images per cluster are randomly selected for annotation. However, for cases with a total of 2, 5, or 10 annotated images, only 1 image per cluster is annotated.

| Dataset | Annotated Images Count | Attributes | | |
|---|---|---|---|---|
| | | Gender(%) | Glasses(%) | Age(%) |
| SG2 (FFHQ) | 2 | 86.28(±13.37) | 83.65(±7.38) | 52.72(±14.49) |
| | 5 | 84.08(±8.48) | 97.05(±5.08) | 70.87(±8.93) |
| | 10 | 85.02(±5.10) | 95.97(±3.88) | 69.90(±6.63) |
| | 30 | 87.35(±1.85) | 97.39(±2.31) | 75.51(±5.83) |
| SG2 (CelebA) | 2 | 89.76(±11.12) | 87.8(±12.97) | 66.93(±15.4) |
| | 5 | 89.16(±7.22) | 91.51(±3.64) | 69.83(±8.98) |
| | 10 | 89.44(±3.95) | 96.56(±2.90) | 70.85(±6.62) |
| | 30 | 91.01(±2.12) | 97.96(±1.39) | 74.48(±5.40) |
| MAE (FFHQ) | 2 | 50.62(±4.30) | 77.06(±8.23) | 54.46(±14.82) |
| | 5 | 51.08(±4.18) | 91.51(±7.44) | 64.67(±11.9) |
| | 10 | 51.71(±3.38) | 90.73(±5.11) | 67.18(±7.37) |
| | 30 | 53.47(±2.83) | 93.39(±3.08) | 72.21(±5.52) |
| MAE (CelebA) | 2 | 50.38(±6.37) | 89.30(±13.1) | 64.98(±17.19) |
| | 5 | 56.36(±6.24) | 89.11(±9.59) | 68.58(±9.62) |
| | 10 | 56.89(±4.64) | 87.84(±6.30) | 66.81(±7.78) |
| | 30 | 58.58(±3.24) | 92.40(±3.04) | 73.15(±4.74) |

## 5.5 JUSTIFICATION OF FEATURE REDUCTION.

**Justification of Few-shot Channel Selection.** To demonstrate the effectiveness of our proposed few-shot channel selection module, we simply compare facial attribute clustering performance with and without this module. The discrepancy column indicates the performance degradation compared to the setup with the few-shot channel selection, as shown in Table 5. For the FFHQ dataset, attribute 'Glasses' remains nearly unaffected, suggesting SG2 model possesses strong 'glasses' extraction capabilities across its overall feature channels. However, significant performance drops are observed for Gender (ACC: -28.54, ARI: -53.98) and Age (e.g., ACC: -12.14, ARI: -35.62), emphasizing the few-shot channel selection's importance for these attributes. Similar results over CelebA further validate its effectiveness.

**Justification of Choice of 4 Images for Each Attribute Class.** As Tab. 4 shows that when the number of images per class exceeds one, the accuracy of few-shot channel selection shows a gradual overall increase, with relatively minor differences. To demonstrate the performance of our method under low human effort, we set 4 images to represent each class.

**Justification of Choice of Max Pooling.** To justify our choice of max pooling (Sec. 4), we compare its performance on few-shot image classification (Sec. 5.4) against mean pooling. As shown in Fig. 8, max pooling consistently outperforms mean pooling, supporting our choice.

Table 3: Few-shot classification accuracy against the total number of annotated images. The annotated images are the same as those in Table 2. True label: the classification accuracy when the full ground truth labels are used in training.

| Dataset | Annotated Images Count | Gender(%) | Glasses(%) | Age(%) |
|---|---|---|---|---|
| | | | Attributes | |
| SG2 (FFHQ) | 2 | 85.88(±11.77) | 92.71(±8.54) | 53.86(±13.93) |
| | 5 | 85.44(±9.15) | 97.54(±3.99) | 73.77(±8.93) |
| | 10 | 88.25(±2.74) | 97.40(±3.06) | 77.09(±4.00) |
| | 30 | 89.36(±1.12) | 98.06(±1.69) | 78.03(±4.36) |
| | True label | 96.47 | 99.04 | 84.58 |
| SG2 (CelebA) | 2 | 90.59(±11.41) | 90.05(±8.30) | 68.05(±16.55) |
| | 5 | 89.23(±9.13) | 95.28(±2.10) | 74.66(±9.44) |
| | 10 | 92.34(±4.44) | 96.89(±1.66) | 76.62(±5.64) |
| | 30 | 93.36(±1.61) | 97.54(±0.91) | 77.56(±0.84) |
| | True label | 97.38 | 98.74 | 89.06 |
| MAE (FFHQ) | 2 | 51.20(±3.78) | 72.95(±17.09) | 50.50(±21.40) |
| | 5 | 50.60(±4.30) | 76.80(±10.82) | 56.81(±17.47) |
| | 10 | 50.42(±4.31) | 77.91(±9.24) | 59.33(±14.51) |
| | 30 | 51.33(±3.97) | 78.08(±8.61) | 62.97(±11.19) |
| | True label | 96.82 | 98.44 | 92.48 |
| MAE (CelebA) | 2 | 50.11(±7.66) | 89.22(±17.87) | 61.94(±22.94) |
| | 5 | 52.83(±7.79) | 90.65(±13.96) | 71.18(±15.37) |
| | 10 | 53.72(±6.88) | 93.54(±1.09) | 71.18(±14.90) |
| | 30 | 54.04(±6.99) | 93.65(±0.13) | 75.98(±7.80) |
| | True label | 98.24 | 93.67(±0.01) | 77.40(±0.42) |

Table 4: Comparison of FFHQ facial attribute clustering accuracy using varying number of images for few-shot channel selection. The number of adopted images for each facial attribute class varying from 1 to 8 are randomly selected. We repeat such selection 50 times and report clustering accuracy and standard deviation.

| Attributes | Number of images for each facial attribute class | | | | |
|---|---|---|---|---|---|
| | 1 | 2-3 | 4-5 | 6-7 | 8 |
| Gender | 58.33±(3.11) | 73.63±(12.73) | 81.04±(10.65) | 84.59±(10.49) | 82.41±(11.77) |
| Age | 67.14±(1.18) | 75.04±(5.30) | 76.55±(4.78) | 77.69±(4.07) | 77.32±(3.75) |
| Glasses | 96.79±(5.32) | 97.72±(0.05) | 97.72±(0.03) | 97.73±(0.03) | 97.74±(0.02) |

Table 5: Compared facial attribute clustering results **without few-shot channel selection** across two datasets (FFHQ and CelebA). The discrepancy highlights the performance degradation when the few-shot channel selection module is removed. Performance drop are highlighted in bold.

| Data Type | Glasses | | | Gender | | | Age | | |
|---|---|---|---|---|---|---|---|---|---|
| | ACC | NMI | ARI | ACC | NMI | ARI | ACC | NMI | ARI |
| SG2(FFHQ) | 97.69 | 80.32 | 91.11 | 59.13 | 2.70 | 3.77 | 66.92 | 0 | 0 |
| Discrepancy | +0.03 | +0.26 | +0.17 | **-28.54** | **-49.81** | **-53.98** | **-12.14** | **-35.48** | **-35.62** |
| SG2(CelebA) | 93.67 | 0 | 0 | 87.43 | 46.52 | 56.03 | 77.47 | 0 | 0 |
| Discrepancy | **-0.4** | **-65.14** | **-80.58** | **-6.9** | **-22.35** | **-22.54** | 0 | 0 | 0 |

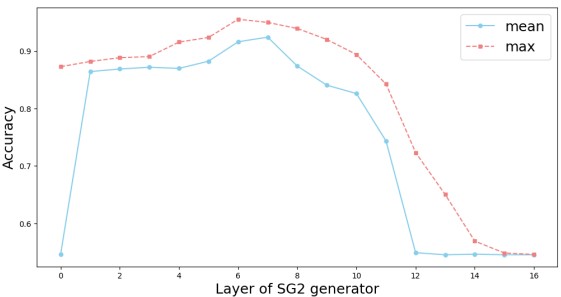

| Model | ACC | NMI | ARI |
|-------|-----|-----|-----|
| SG2 | **98.61** | **97.20** | **97.36** |
| VGG16 | 70.01 | 65.78 | 59.87 |
| MAE | 67.16 | 53.40 | 53.30 |

Figure 9: Clustering results using the representations learned by our SG2-ased method, VGG16, and Masked Autoencoder (MAE) on the AFHQ-Wild dataset. Our method still performs the best on all metrics.

Figure 8: Choice justification of max pooling (Sec. 4).

### 5.6 GENERALIZATION TO NON-HUMAN FACIAL DATASETS

To demonstrate the generalizability of our method to non-human facial datasets, we test our few-shot clustering method (Sec. 5.2) on the AFHQ-Wild dataset . Compared methods are MAE and VGG16. We use ViT-large Dosovitskiy et al. (2020) as the backbone of MAE and pretrain on AFHQ-Wild for 800 epoches, while other settings follow the released code of He et al. (2022). VGG16 is pretrained on ImageNet-1K which contains all 7 species in AFHQ-Wild.

**T-SNE Visualization on AFHQ-Wild.** We also perform t-SNE visualization on the AFHQ-Wild dataset, with results shown in Fig. 10. Similar to the results on human facial datasets, SG2 representations exhibit the clearest distinguishability.

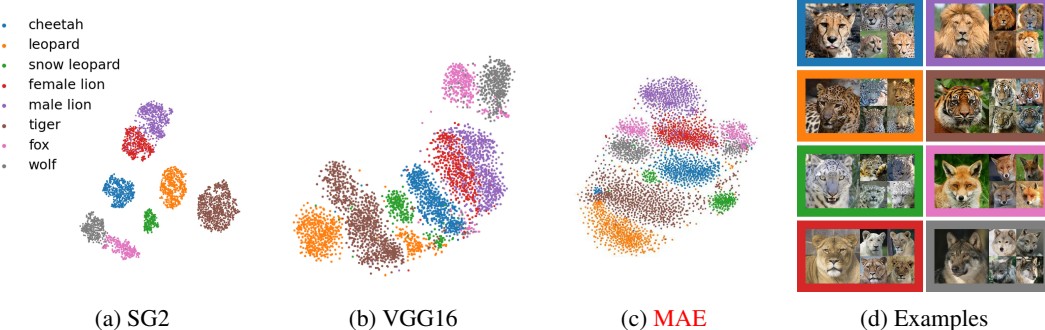

(a) SG2      (b) VGG16      (c) MAE      (d) Examples

Figure 10: Visualization of representations learned by StyleGAN2 (SG2), VGG16, and Masked Autoencoder (MAE) for eight animal classes (cheetah, leopard, snow leopard, female lion, male lion, tiger, fox, wolf) using t-SNE. The results are obtained using 5,000 images from AFHQ-Wild dataset.

## 6 CONCLUSION

In conclusion, we propose a novel unsupervised approach to learn facial attribute representations that leverages the unique capabilities of StyleGAN. By exploiting StyleGAN's Hierarchical Feature Modulation, we have demonstrated a method to automatically discover rich, disentangled representations of facial attributes in an unsupervised manner. The effectiveness of our method is evidenced by its strong performance across a range of unsupervised and few-shot downstream tasks, including facial attribute clustering, few-shot classification, and few-shot facial attribute annotation. To the best of our knowledge, ours is the first method to successfully achieve these results.

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

## A  ADDITIONAL EXPERIMENTAL RESULTS

We show additional experimental results on few-shot annotation and few-shot classification in Table 6 and Table 7, supporting the same conclusions drawn in the main paper.

Table 6: Few-shot annotation accuracy against the total number of annotated images. For most cases, 3 images per cluster are randomly selected for annotation. However, for cases with a total of 2, 5, or 10 annotated images, only 1 image per cluster is annotated.

| Dataset | Annotated Images Count | Attributes | | |
|---|---|---|---|---|
| | | Gender(%) | Glasses(%) | Age(%) |
| SG2 (FFHQ) | 2 | 86.28(±13.37) | 83.65(±7.38) | 52.72(±14.49) |
| | 5 | 84.08(±8.48) | 97.05(±5.08) | 70.87(±8.93) |
| | 10 | 85.02(±5.10) | 95.97(±3.88) | 69.90(±6.63) |
| | 30 | 87.35(±1.85) | 97.39(±2.31) | 75.51(±5.83) |
| | 60 | 89.19(±1.78) | 98.21(±1.24) | 76.76(±3.17) |
| | 120 | 90.03(±1.15) | 98.04(±0.69) | 76.83(±2.32) |
| | 600 | 91.35(±0.64) | 98.47(±0.32) | 80.06(±0.89) |
| SG2 (CelebA) | 2 | 89.76(±11.12) | 87.8(±12.97) | 66.93(±15.4) |
| | 5 | 89.16(±7.22) | 91.51(±3.64) | 69.83(±8.98) |
| | 10 | 89.44(±3.95) | 96.56(±2.90) | 70.85(±6.62) |
| | 30 | 91.01(±2.12) | 97.96(±1.39) | 74.48(±5.40) |
| | 60 | 91.19(±1.52) | 97.31(±1.05) | 75.56(±3.34) |
| | 120 | 93.62(±1.04) | 97.68(±0.56) | 76.46(±2.26) |
| | 600 | 94.58(±0.43) | 97.88(±0.25) | 77.74(±0.81) |
| MAE (FFHQ) | 2 | 50.62(±4.30) | 77.06(±8.23) | 54.46(±14.82) |
| | 5 | 51.08(±4.18) | 91.51(±7.44) | 64.67(±11.9) |
| | 10 | 51.71(±3.38) | 90.73(±5.11) | 67.18(±7.37) |
| | 30 | 53.47(±2.83) | 93.39(±3.08) | 72.21(±5.52) |
| | 60 | 55.07(±2.48) | 94.74(±1.80) | 73.57(±3.94) |
| | 120 | 57.93(±2.02) | 94.49(±1.33) | 74.07(±2.28) |
| | 600 | 62.12(±0.95) | 95.23(±0.45) | 75.31(±1.14) |
| MAE (CelebA) | 2 | 50.38(±6.37) | 89.30(±13.1) | 64.98(±17.19) |
| | 5 | 56.36(±6.24) | 89.11(±9.59) | 68.58(±9.62) |
| | 10 | 56.89(±4.64) | 87.84(±6.30) | 66.81(±7.78) |
| | 30 | 58.58(±3.24) | 92.40(±3.04) | 73.15(±4.74) |
| | 60 | 60.29(±2.84) | 92.27(±2.33) | 72.90(±3.43) |
| | 120 | 62.00(±2.08) | 92.60(±1.42) | 73.37(±2.33) |
| | 600 | 66.77(±0.87) | 92.56(±0.59) | 74.65(±0.92) |

Table 7: Few-shot classification accuracy against the total number of annotated images. The annotated images are the same as those in Table 2. True label: the classification accuracy when the full ground truth labels are used in training.

| Dataset | Annotated Images Count | Attributes | | |
| --- | --- | --- | --- | --- |
| | | Gender(%) | Glasses(%) | Age(%) |
| SG2 (FFHQ) | 2 | 85.88(±11.77) | 92.71(±8.54) | 53.86(±13.93) |
| | 5 | 85.44(±9.15) | 97.54(±3.99) | 73.77(±8.93) |
| | 10 | 88.25(±2.74) | 97.40(±3.06) | 77.09(±4.00) |
| | 30 | 89.36(±1.12) | 98.06(±1.69) | 78.03(±4.36) |
| | 60 | 89.87(±0.69) | 98.41(±0.43) | 78.29(±2.09) |
| | 120 | 93.44(±1.22) | 97.11(±1.13) | 77.59(±0.48) |
| | 600 | 90.03(±0.45) | 98.46(±0.09) | 79.34(±1.03) |
| | True label | | | |
| SG2 (CelebA) | 2 | 90.59(±11.41) | 90.05(±8.30) | 68.05(±16.55) |
| | 5 | 89.23(±9.13) | 95.28(±2.10) | 74.66(±9.44) |
| | 10 | 92.34(±4.44) | 96.89(±1.66) | 76.62(±5.64) |
| | 30 | 93.36(±1.61) | 97.54(±0.91) | 77.56(±0.84) |
| | 60 | 93.44(±1.22) | 97.11(±1.13) | 77.59(±0.48) |
| | 120 | 93.89(±0.79) | 97.11(±1.22) | 77.52(±0.28) |
| | 600 | 94.02(±0.78) | 97.29(±1.03) | 77.50(±0.16) |
| | True label | 97.38 | 98.74 | 88.56 |
| MAE (FFHQ) | 2 | 51.20(±3.78) | 72.95(±17.09) | 50.50(±21.40) |
| | 5 | 50.60(±4.30) | 76.80(±10.82) | 56.81(±17.47) |
| | 10 | 50.42(±4.31) | 77.91(±9.24) | 59.33(±14.51) |
| | 30 | 51.33(±3.97) | 78.08(±8.61) | 62.97(±11.19) |
| | 60 | 52.46(±3.45) | 79.41(±4.16) | 65.22(±6.95) |
| | 120 | 53.27(±2.55) | 78.75(±5.17) | 66.11(±5.56) |
| | 600 | 54.21(±1.73) | 77.03(±12.04) | 64.97(±6.45) |
| | True label | 96.82 | 98.44 | 88.56 |
| MAE (CelebA) | 2 | 50.11(±7.66) | 89.22(±17.87) | 61.94(±22.94) |
| | 5 | 52.83(±7.79) | 90.65(±13.96) | 71.18(±15.37) |
| | 10 | 53.72(±6.88) | 93.54(±1.09) | 71.18(±14.9) |
| | 30 | 54.04(±6.99) | 93.65(±0.13) | 75.98(±7.80) |
| | 60 | 55.61(±5.74) | 93.59(±0.60) | 75.81(±7.28) |
| | 120 | 57.42(±3.53) | 93.65(±0.11) | 76.30(±3.74) |
| | 600 | 58.36(±0.60) | 93.60(±0.65) | 77.03(±2.45) |
| | True label | 98.24 | 99.14 | 88.56 |