# OpenReview forum: "Unsupervised Learning of Facial Attribute Representations Using StyleGAN"
_ICLR.cc/2025/Conference — Submitted to ICLR 2025_

### Official Review · Reviewer_356M · 2024-11-03

**Soundness:** 3
**Presentation:** 3
**Contribution:** 2
**Rating:** 3
**Confidence:** 4

**Summary:**

This study presents an unsupervised learning framework for facial attribute analysis that overcomes the need for extensive annotated datasets. Utilizing StyleGAN, the framework learns rich and disentangled representations of facial attributes without labeled data. The approach capitalizes on StyleGAN's Hierarchical Feature Modulation to automatically identify semantically meaningful features across different layers. To enhance the use of these representations, GAN inversion techniques are employed to convert input images into StyleGAN features, complemented by a feature reduction method based on mutual information. Experiments in few-shot facial attribute analysis, including clustering, classification, and annotation tasks, demonstrate the effectiveness of this framework.

**Strengths:**

The research point is interesting, learning facial attribute representation from an unsupervised perspective.

**Weaknesses:**

1. The novelty of the paper appears limited. The proposed method appears to rely primarily on StyleGAN's feature decoupling capability and does not offer substantial insights.
2. The description of the method is overly brief and difficult to comprehend. The core of the proposed approach is quantitative justification, lacking sufficient explanation of the purpose of the strategy and the definitions of key indicators (IntraMI/InterMI).
3. The main issue with the experimental section is the lack of comparative methods. In fact, there are numerous studies focused on face representation learning that could provide valuable context.
4. Some references focusing on learning facial representations are missing:
[1] General Facial Representation Learning in a Visual-Linguistic Manner
[2] FaceXFormer: A Unified Transformer for Facial Analysis

**Questions:**

Please refer to the Strengths and Weaknesses.

---

> ### Author Response · Authors · 2024-12-01
> **Addressing Concerns: Novel Insights, Methodological Clarity, and Comparative Analysis**
>
> We thank the reviewer for their thoughtful feedback and hope our response addresses the concerns raised.
>
> ### **Q1: StyleGAN's Feature Decoupling Capability and Substantial Insights**
> We respectfully clarify that our work provides significant contributions and meaningful insights by addressing underexplored directions:
>
> **Expanding Beyond Generative Tasks:**
>
> While prior studies predominantly leverage StyleGAN for generative tasks such as style transfer, facial attribute manipulation, and image interpolation, our work shifts the focus toward its use in image representation and facial analysis. Specifically, we explore StyleGAN’s potential as a robust tool for analyzing feature maps, offering new perspectives on its applicability.
> This novel approach broadens the scope of StyleGAN’s utility, transforming it from a generative model into a versatile framework for understanding and interpreting images, thus providing substantial new insights and practical contribution.
>
> **Proposed Methodologies and Insights:**
>
> To substantiate this perspective, we introduce specific methodologies:
>
> * A Mutual Information-based few-shot channel selection method, which effectively identifies discriminative channels within feature maps. This allows for precise feature filtering tailored to specific tasks, enhancing efficiency.
> * A feature reduction strategy, which successfully addresses feature map redundancy and dimensionality reduction, making StyleGAN’s feature maps more interpretable.
> These methods not only demonstrate excellent generalizability across datasets but also facilitate tasks such as analyzing and comparing feature maps.
>
> We believe our work introduces a meaningful and underexplored direction for facial attribute analysis, showcasing the versatility and capability of StyleGAN from a fresh perspective. These contributions provide both theoretical insights and practical value to the field.
>
> ### **Q2: Methodology Description and Clarity of Key Indicators (IntraMI/InterMI)**
> To address concerns about brevity and clarity:
>
> * We have expanded **Section 3 (Quantitative Justification)** to provide detailed explanations of the key indicators, IntraMI and InterMI.
> * Specifically, we now include step-by-step descriptions of how these metrics are computed, along with their relevance in quantifying feature map similarity and separation.
> These revisions aim to enhance the comprehensibility of the methodology and ensure a clear understanding of its purpose and application.
>
> These revisions aim to enhance the comprehensibility of the methodology and ensure a clear understanding of its purpose and application.
>
> ### **Q3: Comparative Experiments with Existing Studies**
> We acknowledge the importance of comparative experiments with existing methods. In our revised manuscript, we supplement additional results in **Figure 7, Table 4 and 5** that contextualize our approach alongside these studies, highlighting the unique advantages of leveraging StyleGAN in unsupervised facial attribute analysis.
>
> ### **Q4: Missing References**
> Thank you for pointing out the missing references. We have revised the Introduction section to include the suggested works:
>
> * [1] General Facial Representation Learning in a Visual-Linguistic Manner.
> * [2] FaceXFormer: A Unified Transformer for Facial Analysis.
>
> These additions enrich the context of our work and position it appropriately within the broader literature.
>
> We hope these revisions adequately address the concerns raised and demonstrate the novelty and contributions of our work.

---

### Official Review · Reviewer_KZux · 2024-11-03

**Soundness:** 2
**Presentation:** 3
**Contribution:** 1
**Rating:** 3
**Confidence:** 5

**Summary:**

This paper aims to learn disentangled and distinguishable features in an unsupervised manner. It employs the inductive bias of StyleGAN to automatically discover semantically meaningful representations of facial attributes

**Strengths:**

Using self-supervised learning to enhance facial feature representation is a reasonable and common approach.

**Weaknesses:**

1.	This article claims that the MAEs-based methods still use a large amount of labeled data. This assertion is questionable. To my knowledge, methods based on MAE can significantly reduce the training data requirements for downstream tasks after pre-training on a large amount of unlabeled data.
2.	The experimental comparison in Fig. 2 is unfair. VGG16 is pre-trained on ImageNet dataset. Comparing to the other model pre-trained on face images, VGG16 naturally performs less well in disentangle the glasses feature. The authors should also pre-train VGG16 on the same FFHQ dataset.
3.	It's strange that this method selects feature channels based solely on 8 images. The paper lacks experimental results to demonstrate that using just 8 images is sufficient to select the appropriate channels.
4.	To my knowledge, this article introduces two uncommon few-shot tasks: clustering and annotation. However, the article does not provide detailed explanations for these tasks nor demonstrate their academic value. Furthermore, this article also lacks comparisons with other state-of-the-art methods on these tasks.
5.	In line 260 and line 261, the sentence “ clustering, annotation, and clustering” is a typo.

**Questions:**

Please address my concerns on the insufficient experiments.

---

> ### Author Response · Authors · 2024-12-01
> **Addressing Reviewer's Concerns on Data Requirements, Model Comparisons, and Methodology in Few-shot Facial Attribute Analysis**
>
> We thank the reviewer for their comprehensive comments and hope our response addresses the concerns raised.
>
> ### **Q1: Data requirements of MAEs for downstream (Unsupervised) task.**
> We agree that MAE is a powerful unsupervised pretraining model for downstream tasks. However, the redundant and unexplicit representation of MAE (150K-dimensional) poses challenges in **unsupervised** downstream tasks. As shown in **Tables 2 and 3**, while MAE demonstrates marginal superiority over classification with the **full set of true labels**, it fails to accomplish few-shot annotation and classification in all three facial attributes.
>
> ### **Q2: Baseline VGG16 on face images is unfair.**
> Thank you for your advice. We acknowledge that as a supervised model pretrained on ImageNet, VGG16 may not serve as a fully convincing comparison against SG2. The primary reason we initially employed VGG16 was its demonstrated effectiveness in decoupling facial attributes. To address this concern, we have now aligned the pretraining settings by replacing VGG16 with an **unsupervised VQVAE model pretrained on FFHQ**. As shown in **Fig. 2**, the updated comparison reveals that VQVAE also struggles to capture disentangled facial attribute 'glasses'.
>
> ### **Q3: Lack of ablation study of the number of images for few-shot channel selection.**
> **Table 4** shows the ablation study regarding the number of images used for few-shot channel selection. It justifies that **4 images** per facial class are sufficient for performing few-shot channel selection, thus minimizing human effort.
>
> ### **Q4: Academic value of few-shot clustering and classification? Lack of comparison with SOTA methods.**
> We compare the few-shot facial attribute classification of our method against the powerful supervised model **ResNet-18** in **Fig. 7**. It can be observed that in a supervised setting, ResNet-18 requires more than 400 images to achieve comparable results as SG2's '2-shot' classification accuracy. This showcases the challenges of facial attribute analysis in an unsupervised setting.

---

### Official Review · Reviewer_SUxr · 2024-11-03

**Soundness:** 2
**Presentation:** 2
**Contribution:** 2
**Rating:** 3
**Confidence:** 3

**Summary:**

This work proposes to employ an unsupervised approach for facial attribute analysis, eliminating reliance on large labeled datasets. It illustrates that StyleGAN has a superior inductive bias, enabling the extraction of disentangled and interpretable facial attribute features at different layers. And A mutual information-based feature refinement method is proposed to enhance the effectiveness of these features in downstream unsupervised tasks.

**Strengths:**

1. This paper introduces the unsupervised approach to facial attribute analysis, reducing dependence on labeled data by fully exploring StyleGAN's feature extraction capabilities. It also introduces a novel mutual information-based feature refinement method, enhancing representation quality for unsupervised tasks.
2. The authors provide experiments across four classification tasks in three scenarios to demonstrate the framework's effectiveness.

**Weaknesses:**

1. Necessity of unsupervised facial attribute classification task

The primary contribution of the paper is its unsupervised approach to facial attribute classification, yet the necessity of this task not sufficiently justified:

a) Data Requirements for Supervised Classification: For relatively straightforward binary and ternary classification tasks (e.g., gender, presence of glasses, or age group), it is unclear if large amounts of labeled data are indeed required to achieve satisfactory results in a supervised setting. To establish the necessity of an unsupervised approach, the authors should provide experiments quantifying the data requirements for supervised classifiers on these tasks. For example, showing performance levels and data demands for a standard supervised classifier could substantiate the need for an unsupervised approach.

b) Definition and Challenge of the Facial Attribute Analysis Task: The paper's definition of facial attribute analysis is somewhat ambiguous, as it appears to center only on simple binary and ternary classification tasks. Additionally, in the related work section, the cited literature does not treat facial attribute analysis as a primary task, often addressing it as an auxiliary component. Moreover, the specific attributes addressed here do not closely align with those in the cited works. Therefore, the authors should clarify the precise scope and significance of this task and its challenges in a way that more directly addresses the novelty of this contribution. Alternatively, citing literature that better aligns with the defined problem scope would strengthen the paper.


2. Insufficient explanation and experimentation on StyleGAN feature superiority
While the paper claims that StyleGAN features provide superior performance, further clarification and experimentation are needed to substantiate this claim:

a) Performance in Few-Shot Classification with True Labels: In the unsupervised few-shot classification setting presented in Table 3, when labels are set as TrueLabel, the MAE model achieves better results than StyleGAN (SG2) on certain tasks. The authors should explain why MAE outperforms SG2 here and clarify the contexts or conditions in which StyleGAN features are expected to excel over alternative approaches like MAE.

b) Feature Reduction Method Justification: When discussing the feature reduction approach, the experiments only compare max pooling with mean pooling, without establishing the necessity of the reduction operation itself. To substantiate the proposed feature reduction, the authors should provide experiments demonstrating the effectiveness of reduction as a whole, perhaps by comparing results with and without reduction to highlight its impact on the performance.

c) Generalization Claims of StyleGAN Features: The generalization capability of StyleGAN features is tested using the AFHQ-Wild dataset. However, since StyleGAN was pre-trained on this dataset, while MAE and VGG were not, this setup does not offer a fair assessment of StyleGAN’s generalization capabilities. To strengthen this claim, additional experiments should be conducted on datasets not used in StyleGAN's pre-training, thereby providing a more unbiased comparison of generalization between these models.

**Questions:**

1. Necessity, Definition, and Challenges of the Task

Please provide experiments or analyses showing data requirements for simple supervised classification tasks (e.g., gender, age group) to demonstrate the necessity of an unsupervised approach. Clarifying the specific scope and unique challenges of facial attribute analysis would also help establish its importance and relevance in this work.

2. Clarifications and Experiments on StyleGAN Feature Superiority

Could you clarify why MAE outperforms SG2 in the few-shot classification with True Labels? Additionally, please provide justification for the feature reduction step itself (beyond comparing max and mean pooling) and consider testing StyleGAN’s generalization on datasets it hasn’t been pre-trained on to offer a more balanced view.

---

> ### Author Response · Authors · 2024-12-01
> **Data Requirements, StyleGAN Feature Superiority, and Generalization across Datasets**
>
> We thank the reviewer for their insightful comments and have addressed the concerns as follows.
>
> ### **Q1: Data Requirements for Supervised Facial Attribute Classification (Challenges of the Task).**
>
> **Figure 7** demonstrates that in a supervised setting, **ResNet-18** requires over 400 labeled images to achieve a classification accuracy comparable to StyleGAN2's performance with only 2-shot annotation. This highlights the **data-intensive nature** of supervised facial attribute classification and underscores the necessity of developing unsupervised approaches to handle tasks with limited labeled data.
>
> ### **Q2: Justification of Feature Reduction Method.**
>
> We appreciate the reviewer’s feedback and have provided additional results to address this concern. In the revised manuscript **(Section 5.5)**, we have included **Table 5**, which demonstrates the effectiveness of our feature reduction method by evaluating its impact on facial attribute clustering performance, comparing results with and without the use of few-shot channel selection.
>
> Furthermore, **Table 4** provides an ablation study validating the choice of using 4 images per attribute class for few-shot channel selection. These results collectively highlight the necessity and impact of the proposed feature reduction approach in enhancing task-specific performance while minimizing redundancy.
>
> ### **Q3: Clarifications and Experiments on StyleGAN Feature Superiority**
>
> * **MAE vs. SG2 in Few-Shot Classification with True Labels.**
> Thank you for highlighting this point. The MAE's superiority can be explained by the context in which this result was obtained. Specifically, this was a **fully supervised setting**, rather than a genuine few-shot classification scenario.
> MAE’s advantage can be attributed to its high-dimensional representation space. This characteristic simplifies binary and ternary classification tasks by offering a finer granularity of features that align well with supervised learning objectives.
> In contrast, SG2 excels in **unsupervised settings**, particularly in tasks requiring **feature disentanglement** and generalization with minimal labeled data. Its architecture and feature decoupling capabilities are optimized for scenarios where data or annotations are limited, as demonstrated in the few-shot annotation and classification experiments in **Tables 2 and 3**.
>
> * **Unfair setup on AFHQ-Wild Comparison.**
> We appreciate the reviewer’s observation regarding the pretraining setup. To ensure fairness and alignment, we have updated our experimental setup. Specifically, we **pretrain MAE with a ViT-Large backbone on AFHQ-Wild for 800 epochs**. For VGG16, we note that it was pretrained on ImageNet-1K, a dataset that **already** includes the 7 species present in AFHQ-Wild. The updated comparison results, shown in Figure 10, clearly indicate that StyleGAN2 continues to demonstrate superior representation capabilities compared to these models, even under this revised experimental setup.
>
> * **StyleGAN’s Generalization Across Datasets.**
> To further evaluate StyleGAN2's generalization capabilities beyond the AFHQ-Wild dataset, we evaluated StyleGAN2's generalization capability by testing it on the **CelebA** dataset, even though it was pretrained on FFHQ dataset. The results of this cross-dataset evaluation, presented in **Tables 2 and 3** highlights StyleGAN2's strong generalization performance in few-shot annotation and classification tasks. These findings effectively address concerns about StyleGAN2’s generalization and validate its robustness in handling datasets it was not pretrained on.
>
> We hope these clarifications and additional insights address the reviewer’s concerns and emphasize the relevance and robustness of our approach.

---

### Meta-Review · Area_Chair_jxG6 · 2024-12-17

**Metareview:**

This paper aims to learn disentangled and distinguishable features in an unsupervised manner. The authors propose an unsupervised method for facial attribute analysis to eliminate reliance on large labeled datasets. Experiments are performed to evaluate the effectiveness of the proposed method. Reviewers raised concerns about the task setup, experiential result, logic and presentation of the methodology. Based on the above considerations, I think the current manuscript does not match the ICLR’s requirement and I do not recommend to accept this manuscript.

**Additional Comments On Reviewer Discussion:**

Three reviewers gave consistent negative rating scores. Although authors provided rebuttals for each reviewer, the responses have not sufficiently addressed the reviewers’ concerns.

---

### Decision · Program_Chairs · 2025-01-22

Reject